# Plant Responses to Heavy Metal Stresses: Mechanisms, Defense Strategies, and Nanoparticle-Assisted Remediation

**DOI:** 10.3390/plants14243834

**Published:** 2025-12-16

**Authors:** Aysha Siddika Jarin, Md Arifur Rahman Khan, Tasfiqure Amin Apon, Md Ashraful Islam, Al Rahat, Munny Akter, Touhidur Rahman Anik, Huong Mai Nguyen, Thuong Thi Nguyen, Chien Van Ha, Lam-Son Phan Tran

**Affiliations:** 1Department of Agronomy, Gazipur Agricultural University, Gazipur 1706, Bangladesh; ayshabsmrau@gmail.com (A.S.J.); munny@gau.edu.bd (M.A.); 2Institute of Genomics for Crop Abiotic Stress Tolerance, Department of Plant and Soil Science, Texas Tech University, Lubbock, TX 79409, USA; tanik@ttu.edu (T.R.A.); huong.nguyen@ttu.edu (H.M.N.); 3Agronomy and Farming Systems Division, Bangladesh Sugarcrop Research Institute, Ishwardi 6620, Bangladesh; aponbsmrau11@bsri.gov.bd; 4Department of Genetics and Plant Breeding, Gazipur Agricultural University, Gazipur 1706, Bangladesh; ashraful4225@stu.gau.edu.bd; 5Department of Agroforestry, Bangladesh Agricultural University, Mymensingh 2202, Bangladesh; alrahat.bsmrau@gmail.com; 6Vietnam National University of Agriculture, Ha Noi 100000, Vietnam; ntthuong355@gmail.com

**Keywords:** heavy metal stress, nanoparticles, nanoparticle-assisted remediation, oxidative stress, phytotoxicity, signaling pathways

## Abstract

Heavy metal (HM) contamination threatens environmental sustainability, food safety, and agricultural productivity worldwide. HM toxicity adversely affects plant growth, reducing germination rates by 20–50%, impairing seedling establishment, and inhibiting shoot and root development by 30–60% in various crops. HM disrupts key physiological processes, including photosynthesis, stomatal regulation, membrane integrity, nutrient uptake, and enzymatic and nonenzymatic antioxidant activities. These disruptions largely result from oxidative stress, caused by the excessive accumulation of reactive oxygen species, which damage cellular components. To counteract HM toxicity, plants deploy a complex defense network involving antioxidant enzymes, metal chelation by phytochelatins and metallothioneins, vacuolar sequestration, and symbiotic interactions with arbuscular mycorrhizal fungi, which can retain 40–70% of metals in roots and reduce translocation to shoots. At the molecular level, MAPK (Mitogen-Activated Protein Kinase) signaling pathways, transcription factors (e.g., WRKY, MYB, bZIP, and NAC), and phytohormonal crosstalk regulate the expression of stress-responsive genes expression to enhance HM stress tolerance. Advances in nanotechnology offer promising strategies for the remediation of HM-contaminated soils and water sources (HM remediation); engineered and biogenic nanoparticles (e.g., ZnO, Fe_3_O_4_) improve metal immobilization, reduce bioavailability, and enhance plant growth by 15–35% under HM stresses, although excessive doses may induce phytotoxicity. Future applications of nanotechnology in HM remediation should consider nanoparticle transformation (e.g., dissolution and agglomeration) and environmentally relevant concentrations to ensure efficacy and minimize phytotoxicity. Integrating phytoremediation with nanoparticle-enabled strategies provides a sustainable approach for HM remediation. This review emphasizes the need for a multidisciplinary framework linking plant science, biotechnology, and nanoscience to advance HM remediation and safeguard agricultural productivity.

## 1. Introduction

Heavy metals (HMs) are dense metallic elements with densities exceeding 5 g cm−^3^ and relatively high atomic weights [1,2,3]. HMs are non-biodegradable and can persist in the environment for long periods, leading to prolonged biological half-lives [1,2,3]. Arsenic (As), cadmium (Cd), chromium (Cr), mercury (Hg), and lead (Pb) are common HMs found in soils, water, and wastewater, primarily introduced via natural processes, anthropogenic activities, and agricultural practices [4,5,6,7]. In the rhizosphere, HMs interact with root exudates, altering their solubility and bioavailability, which may impair nutrient uptake and induce mineral deficiencies in plants [8,9]. HMs disrupt physiological and biochemical processes by generating oxidative stress, impairing protein synthesis and cell division, and causing symptoms such as leaf chlorosis and growth inhibition [9,10,11,12,13,14,15,16,17,18,19]. Cellular components, including plasma membranes, mitochondria, and chloroplasts are particularly vulnerable to HM-induced toxicity [15,20]. To cope with HM stress, plants have evolved several adaptive mechanisms, such as regulating metal uptake, enhancing antioxidant defenses, chelating and compartmentalizing HMs, and activating stress-responsive signaling pathways [8,9,21,22]. Detoxification strategies include symbiotic associations with mycorrhizal fungi, immobilization of metal ions via root exudates or cell wall binding, active efflux of metals across membranes, chelation by metal-binding peptides such as phytochelatins and metallothioneins, and sequestration into vacuoles to prevent cytosolic toxicity [23,24,25]. Phytoremediation represents a sustainable strategy for mitigating HM contamination, utilizing the metal-accumulating capabilities of plants and their associated rhizosphere microorganisms [24,25]. Furthermore, the incorporation of nanotechnology into remediation strategies also shows potential to enhance HM removal efficiency. However, its effectiveness depends on the type of metal, soil characteristics, and nanoparticle properties [8,26,27]. To fully harness the potential of nanoparticle-assisted remediation, a comprehensive understanding of the nanoscale interactions and the associated cellular and molecular mechanisms is essential.

This review aims to explore the morphological, physiological, biochemical, and molecular responses of plants to HM stress, focusing on their underlying defense mechanisms and current strategies in phytoremediation and nanoparticle-assisted remediation. Furthermore, it highlights critical knowledge gaps and outlines future research directions for the effective management of HM contamination.

## 2. Roles of Heavy Metals in Plants

HM stress negatively affects plant growth and development [28,29]. Table 1 presents the roles as well as the adequate and toxic levels of heavy metals in plants. While some HMs serve as essential micronutrients at trace levels, their excessive accumulation in plant tissues causes toxicity [30,31,32]. The severity of HM toxicity depends on the specific metal species, its chemical form, concentration, and plant type [33,34]. Non-essential HMs, such as As (≥5 mg kg^−1^ dry weight, DW), Cd (≥5 mg kg^−1^ DW), Cr (≥5 mg kg^−1^ DW), Hg (≥1 mg kg^−1^ DW), and Pb (≥30 mg kg^−1^ DW) are toxic to plants (Table 1). For example, Cd accumulation at toxic levels disrupts water and nutrient uptake, induces oxidative stress, impairs root development, and ultimately reduces crop productivity and food safety [35,36]. Hg predominantly accumulates in roots, with possible translocation to shoots, where it interferes with photosynthesis, respiration, and overall cellular metabolism by generating reactive oxygen species (ROS) and damaging cellular membranes [37,38]. Cr inhibits seed germination by 20–50%, reduces amylase activity, alters sugar transport to embryos, and decreases photosynthetic efficiency through oxidative damage [39,40]. Pb interferes with root and shoot growth (30–60%) in various crops as well as chlorophyll synthesis, stomatal regulation, and photosystem II efficiency, thereby impairing CO_2_ assimilation [41,42]. As, particularly arsenite (As^3+^, AsO_3_^3–^), is primarily taken up via aquaporin channels and binds to sulfhydryl groups in proteins, inhibiting respiration by targeting dehydrogenases, promoting ROS production, and causing intrachromosomal recombination. In contrast, arsenate (As^5+^, AsO_4_^3–^) is taken up mainly through phosphate transporters and is largely reduced to As^3+^ in roots, resulting in limited translocation to shoots, whereas As^3+^ can translocate more readily depending on the plant species and metal detoxification mechanisms [43]. Furthermore, combined exposure to HMs can exacerbate leaf and stem wilting, in purple loosestrife (*Lythrum salicaria*) and maize (*Zea mays*) [44,45]. Essential micronutrients, including copper (Cu), iron (Fe), manganese (Mn), molybdenum (Mo), and zinc (Zn), are vital for optimal vegetative and reproductive development in plants. Both deficiencies and excesses of these elements can disrupt physiological and biochemical processes, resulting in impaired growth, reduced photosynthetic efficiency, nutrient imbalances, and decreased crop productivity [46,47]. For instance, Cu (5–30 mg kg^−1^ DW) and Zn (25–150 mg kg^−1^ DW) concentrations enhance enzyme activity, photosynthetic efficiency, and antioxidant defense systems, thereby supporting overall plant growth and productivity [46,47,48,49]. Additionally, Cu and Zn nanoparticles have been reported to promote seed germination, root elongation, and stress tolerance by modulating ROS homeostasis and enhancing nutrient uptake [46,47,48,49,50,51]. Excess Cu (>30 mg kg^−1^ DW) inhibits seed germination and sugar metabolism by suppressing α-amylase and invertase activities, resulting in growth retardation and leaf chlorosis [51]. Similarly, excessive Fe (>500 mg kg^−1^ DW) and Mn (>1000 mg kg^−1^ DW) promote ROS generation, leading to membrane, protein, and DNA damage, which manifests as bronzing, chlorosis, and inhibited plant growth [52,53]. Elevated Mo interferes with nitrogen fixation in legumes, resulting in stunted growth and floral abnormalities [54]. High Zn concentrations (>150 mg kg^−1^ DW) suppress enzymatic activities (α-amylase, protease, ribonuclease), impair the mobilization of food reserves and carbonic anhydrase activity, increase malondialdehyde (MDA) content and electrolyte leakage, and ultimately reduce plant biomass [55,56,57,58,59]. Mung bean (*Vigna radiata*) and tomato (*Solanum lycopersicum*) exhibit enhanced growth at cobalt (Co) 0.02–2 mg kg^−1^ DW levels, whereas concentrations of >10 mg kg^−1^ DW induce toxicity, causing stunted growth and physiological disturbances [49,50]. Figure 1 illustrates the varying impacts of HMs on plants at toxic concentrations.

## 3. Morphological, Physiological, and Biochemical Responses to Heavy Metal Stresses

HM stress adversely affects morphological, physiological, and biochemical processes in plants. An overview of HM-induced changes in plant morphological, physiological, and biochemical processes is provided in Table 2. One of the primary effects of HM stress is a reduction in intra-leaf CO_2_ levels, which refers to the concentration of CO_2_ within the leaf mesophyll and directly affects photosynthetic carbon assimilation and leaf gas exchange characteristics [71,72]. HM stress adversely affects stomatal function through both direct accumulation in leaf tissues and the indirect disruption of key physiological processes, ultimately reducing the rates of photosynthesis and transpiration [72,73,74]. The regulation of ionic fluxes within guard cells plays a pivotal role in HM-induced stomatal dysfunction [73,74]. Since stomatal conductance directly influences CO_2_ uptake and water vapor loss, any alteration in stomatal morphology such as changes in size, density, or opening behavior can significantly impact leaf gas exchange, nutrient acquisition, and plant water balance [74]. For instance, Pb accumulation in soybean (*Glycine max*) guard cells disrupts cytoplasmic membrane integrity, resulting in potassium ion efflux, reduced guard cell turgor, and stomatal closure [75]. Additionally, Pb stress reduces transpiration in tobacco (*Nicotiana tabacum*) [76] and stomatal conductance in wheat (*Triticum aestivum*) and broadleaf plantain (*Plantago major*) [77,78]. Cd exposure induces stomatal closure, leading to decreased rates of photosynthesis, respiration, and water use efficiency, ultimately disrupting the plant’s energy [79,80]. In wheat, Cd stress results in lowered CO_2_ assimilation, reduced stomatal conductance, and diminished water use efficiency [81]. Cr stress significantly reduces stomatal conductance in maize (*Zea mays*), thereby limiting leaf gas exchange and negatively affecting photosynthetic activity [82]. Similarly, Zn stress reduces stomatal conductance in pigeon pea (*Cajanus cajan*) [83].

Plants counteract HM toxicity through the synthesis of phytochelatins (PCs) and metallothioneins (MTs), which chelate metals and reduce cytosolic toxicity. PCs preferentially bind Cd^2+^ and As^3+^, forming stable vacuolar complexes that limit cytosolic accumulation and oxidative damage, while MTs exhibit variable metal-binding affinities depending on the plant species. The specificity of PCs and MTs influences detoxification efficiency; for example, Cd^2+^ is rapidly chelated in roots, reducing shoot translocation, whereas As^3+^ complexes may be redistributed gradually. These mechanisms vary across species, with Arabidopsis (*Arabidopsis thaliana*) accumulating Cd–PC complexes mainly in roots, while rice (*Oryza sativa*) shows greater root-to-shoot Cd translocation. Differences in MT expression further modulate tolerance to multi-metal stress, highlighting species-specific strategies in HM detoxification. HM stress increases abscisic acid (ABA) synthesis, which activates the OST1 (Open Stomata 1) protein kinase while inhibiting protein phosphatase 2C (PP2C) activity [96,97]. Blocking PP2Cs prevents protein dephosphorylation, thus relieving the suppression of ABA signaling [98]. The Open Stomata 1 (OST1) represents one of the main targets of PP2Cs, with OST1 being the best-characterized member [100,101]. OST1 transmits the ABA signal by phosphorylating downstream targets and enhancing ROS production via NADPH oxidase activation [102]. Additionally, nitric oxide (NO) and reactive nitrogen species (RNS) act as secondary messengers, amplifying the signaling cascade under HM stress. As a redox-active free radical, NO can neutralize HM-induced ROS directly or by activating antioxidant defenses. However, an imbalance between NO, ROS, and antioxidant activity can trigger oxidative stress [103]. HM stress also disrupts the photosynthetic electron transport chain, leading to decreased Photosystem II activity. At elevated concentrations, HM stress disturbs the light-harvesting complex, particularly the transition between its functional states. This results in antenna complex disorganization and compromised photochemical efficiency [104]. HM stress disrupts glucose metabolism, tricarboxylic acid cycle, and photosynthesis, while inducing ABA, ROS, and NO production, leading to signaling cascades including PP2C, OST1, and 8-Nitro-cGMP that cause turgor loss and stomatal closure. These processes collectively reduce the rates of photosynthesis, transpiration, respiration, nutrient and water uptake, and leaf area, while causing ion imbalance and cell membrane damage. The impact of HM stress on the physiological functions of plants is illustrated in Figure 2.

## 4. Disruption of Cell Membrane Integrity, Oxidative Homeostasis, and Enzymatic Activities Under Heavy Metal Stresses

HMs can penetrate cell membranes and bind to essential components such as proteins and phospholipids [105]. HMs interfere with membrane integrity, disrupt transport processes, displace calcium ions from critical binding sites, and reduce the availability of ATPase substrates by binding to adenosine triphosphate (ATP) [106,107]. HM stress strongly promotes the formation of hydroxyl free radicals, triggering lipid peroxidation through primary initiation as well as via the Haber–Weiss and Fenton reactions [108]. HM stress also leads to the accumulation of methylglyoxal in plants, which impairs antioxidant defense by depleting glutathione levels [109,110,111]. Redox-active metals such as Cu, Fe, and Cr can directly generate ROS [112], while redox-inactive metals such as Cd, Pb, and Zn promote ROS accumulation indirectly by impairing the plant’s antioxidant defense system, triggering calcium-dependent signaling pathways, and disturbing iron metabolism [111,112,113,114,115,116]. On the other hand, Cd, Pb, Ni, and Cr can substitute for vital cofactors like Mg, Zn, Fe, and Ca in enzymes such as RuBisCO and superoxide dismutases, resulting in impaired enzymatic activity and disrupted cellular signaling [117,118]. For example, Cd disrupts cellular functions by displacing Zn in zinc-finger transcription factors and Ca in calmodulin, thereby altering gene expression and impairing stomatal regulation, whereas Pb interferes with chlorophyll structure and photosystem II activity, ultimately reducing photosynthetic efficiency [119,120]. Additionally, arsenate (AsO_4_^3–^) stress can mimic phosphate and disrupt ATP synthesis [121]. It also binds strongly to protein sulfhydryl (–SH) and carboxyl (–COOH) groups, leading to enzyme denaturation [105,122,123,124,125,126,127].

## 5. Plants’ Resistance Mechanisms to Heavy Metal Stresses

Plants cope with HM stress through two main resistance strategies: avoidance and tolerance. Avoidance mechanisms, such as mycorrhizal immobilization, root exudate complexation, and rhizosphere pH adjustment, function to prevent metal uptake and entry into root tissues. Among these avoidance strategies, immobilization by mycorrhizal associations is particularly effective for its role in restricting metal translocation and mitigating toxicity. When metals do enter plant cells, tolerance mechanisms are triggered, including the enhancement of antioxidant defenses, production of stress-associated proteins, and gene regulation to detoxify and compartmentalize the metals into less harmful forms in plants.

### 5.1. Immobilization by Mycorrhizal Associations

Mycorrhizal fungi, particularly arbuscular mycorrhizal fungi (AMF) and ectomycorrhizal fungi (ECM), play a vital role in enhancing plant tolerance to HM stress by forming symbiotic associations with roots [128,129]. These fungi reduce HM uptake through chelation, adsorption, and the release of organic acids and glomalin [130,131]. Extraradical hyphae of AMF extend beyond the root zone, improving nutrients such as N, P, and K uptake, which in turn enhances plant growth and HM stress resistance [132,133]. AMF can retain 40–70% of metals in root tissues via chelation, while extraradical and intraradical hyphae provide additional binding surfaces containing glucan, chitin, and galactosamine polymers [134]. They also release organic acids such as citric, malic, and oxalic acids that immobilize HMs by forming stable complexes [131]. A major contribution of AMF is the secretion of glomalin, a glycoprotein with high metal-binding affinity, which stabilizes soil aggregates and reduces HM bioavailability. Evidence shows that AMF inoculation promotes HM retention in roots and reduces their movement to aerial tissues, often resulting in significantly lower concentrations in edible parts [135]. For example, Adeyemi et al. [134] reported that AMF inoculation enhanced P uptake and restricted HM transport to shoots in soybean (*Glycine max*), while eggplants (*Solanum melongena*) inoculated with AMF accumulated less Pb in fruits, improving growth and food safety. Although most studies confirm reduced translocation, small amounts of metals may still reach aboveground tissues at lower concentrations, and the precise mechanisms controlling long-distance movement remain only partly understood. In addition to AMF, ECM form a fungal sheath around roots and secrete extracellular polymers and organic acids that immobilize metals in the rhizosphere. ECM are particularly important for woody plants and act primarily as barriers to HM entry, whereas AMF combine nutrient acquisition with HM detoxification. Together, these two groups of fungi represent complementary strategies for enhancing plant tolerance to metal-contaminated soils. Through these mechanisms, mycorrhizal associations enhance nutrient transport, detoxify HMs, and improve biomass accumulation, especially during active phases of plant growth. They also block pathogen invasion, enrich beneficial rhizosphere microbes, and modulate phytohormone levels (e.g., IAA, cytokinins), further supporting root health and plant vigor [136,137,138,139]. Figure 3 illustrates the enhancement of HM detoxification, nutrient transport, and biomass accumulation in plants through microbial functions. Thus, mycorrhizal symbiosis functions as a critical plant defense strategy in HM-contaminated environments, providing both direct detoxification and indirect growth benefits. Nonetheless, knowledge gaps remain regarding the regulation of HM translocation from roots to shoots, warranting further research to fully elucidate these processes.

### 5.2. Root-Mediated Mechanisms and Phytochelatin-Driven Detoxification Under Heavy Metal Stresses

Among the diverse strategies employed by plants to withstand HM toxicity, root-mediated mechanisms and PC-driven detoxification play pivotal roles. Cellular mechanisms of HM detoxification and sequestration in plants are illustrated in Figure 4. Roots serve as the primary site for HM perception and uptake. One of the key defense strategies involves the secretion of root exudates, which include diffusates (e.g., peptides, carboxylic acids, and carbohydrates), excretions (e.g., protons, bicarbonates, and CO_2_), and secretions (e.g., mucilage, siderophores, and allelochemicals) [140]. These compounds contribute to the formation of stable metal complexes in the rhizosphere, thereby reducing HM solubility, mobility, and toxicity near the root surface [141]. Organic acids chelate metal ions and reduce their toxicity. Additionally, amino acids like histidine and nicotianamine act as metal chelators, enhancing the solubility and translocation of micronutrients such as Fe and Zn. Root-derived sugars and amino acids can also increase rhizosphere pH and modify redox conditions, encouraging metal precipitation and improving nutrient availability [142]. PCs and MTs play critical roles in chelating metal ions and mediating their compartmentalization particularly into vacuoles thereby minimizing cytosolic toxicity [143,144,145,146]. PCs can form stable complexes with a wide range of metal ions including Cd^2+^, As^3+^, Pb^2+^, and Zn^2+^ [147]. Their synthesis is rapidly upregulated following HM exposure, utilizing cytosolic glutathione present in millimolar concentrations as a substrate [148]. Once formed, PC-metal complexes are actively transported into the vacuole via tonoplast-localized ATP-Binding cassette transporters or metal/H^+^ antiporters, effectively sequestering toxic metals from the cytosol and protecting sensitive enzymatic processes [145]. Inside the vacuole, these complexes may undergo further stabilization through interactions with inorganic sulfide or sulfite ions, which shield them from proteolytic degradation. Over time, this compartmentalization may result in the formation of insoluble crystalline metal deposits, contributing to long-term immobilization and detoxification of HMs [148]. In addition to their detoxification function, PCs are also involved in metal homeostasis, particularly in regulating the uptake and distribution of essential micronutrients such as Zn and Cu [149,150]. In parallel, MTs serve as metal-binding ligands that assist in detoxifying excess metal ions while also regulating the availability of essential elements such as Zn and Cu. In plants, MTs are categorized into subfamilies (e.g., MT1 and MT2) based on the arrangement of their cysteine domains, which determine their metal-binding specificity and biological roles [151]. In addition to PC- and MT-mediated chelation, plants utilize several cellular barriers and strategies to limit HMs entry and toxicity:i.Plant’s cell wall, rich in pectin and lignin, can adsorb HMs through ion exchange, thereby reducing metal mobility and entry into cells [152].ii.Specific membrane transporters and channel proteins regulate HM uptake. Under HM stress, plants often downregulate these transporters to reduce metal influx [118].iii.Chelation is a widespread detoxification mechanism involving intracellular ligands such as PCs, MTs, organic acids (e.g., citric, malic, oxalic acids), and amino acids (e.g., histidine, nicotianamine), which bind free metal ions and prevent them from interfering with vital metabolic processes [153].iv.The tonoplast (vacuolar membrane) minimizes HM movement back into the cytoplasm via active permeability mechanisms, serving as a dynamic barrier against metal recirculation [147].v.Once PC-metal complexes are formed, they are transported into vacuoles and ultimately converted into insoluble metal deposits, stabilizing the intracellular environment and preventing oxidative damage [149,153].

### 5.3. Antioxidant Responses to Heavy Metal Stresses

Antioxidant enzymes play a central role in mitigating oxidative damage in plants exposed to HM stresses [112,113,114]. HM exposure induces the production of heat shock proteins (HSPs), which act as molecular chaperones, helping stabilize proteins and preserve membrane integrity. Studies show that various HSPs, including low-molecular-weight forms and HSP70, are upregulated in different plant species under HM stress [154,155,156,157]. Cd, Pb, Cr, and Hg induce overproduction of ROS, including superoxide radicals, hydrogen peroxide, and hydroxyl radicals, which disrupt cellular homeostasis [158,159]. In response, plants activate enzymatic defense mechanisms involving superoxide dismutase (SOD), catalase (CAT), peroxidases (POD), ascorbate peroxidase (APX), and glutathione reductase (GR) [160,161,162,163,164]. However, the intensity and efficiency of these responses vary with plant species, developmental stage, metal type, and exposure level. An overview of plant antioxidant enzyme responses to HM stress is provided in Table 3. In addition to enzymatic antioxidants, plants also accumulate secondary metabolites which further support ROS detoxification and improve stress resilience [161,165,166,167,168]. These metabolites help mitigate HM toxicity by chelating metals in the rhizosphere and scavenging ROS within the cells.

### 5.4. Regulation of Signaling Pathways and Gene Expression in Plants Under Heavy Metal Stresses

HM exposure triggers complex signaling pathways in plants that regulate gene expression and activate defense mechanisms. A major downstream signaling route is the mitogen-activated protein kinase cascade, which integrates signals from ROS, nitric oxide, and phytohormones to regulate stress-responsive genes [177]. These coordinated responses help initiate detoxification and stress adaptation processes. Crosstalk among signaling pathways ensures fine-tuned regulation, enabling plants to maintain metal homeostasis and resist toxicity [178]. Key signaling pathways and associated molecular players implicated in plant adaptation to HM stress are summarized in Table 4. Understanding these mechanisms is essential for advancing crop resilience to HM-contaminated environments.

Transcription factors (TFs) play a pivotal role in regulating plant responses to HM stress by modulating the expression of stress-responsive genes [185,186,187]. An overview of transcription factors activated during HM stress in plants is provided in Table 5. TFs act as molecular switches that activate or repress target genes involved in metal uptake, transport, sequestration, and detoxification. Key TF families, such as bZIP (basic leucine zipper transcription factor), MYB (myeloblastosis transcription factor), WRKY (tryptophan–arginine–lysine–tyrosine transcription factor), NAC, and AP2/ERF, coordinate a wide range of physiological and biochemical processes, including antioxidant defense, synthesis of metal-binding ligands (e.g., PCs and MTs), and regulation of ion transporters [187,188,189,190,191,192,193]. Mechanistically, WRKY TFs are frequently upregulated under Cd stress, where they activate downstream genes encoding antioxidant enzymes and phytochelatin synthases, thereby enhancing detoxification capacity. MYB TFs preferentially respond to Pb stress, regulating secondary metabolism and ion transport to limit Pb translocation to shoots [182,194,195]. bZIP TFs are strongly linked to redox homeostasis, integrating ROS signaling with metal-induced oxidative stress responses, while NAC and AP2/ERF TFs coordinate hormonal signaling pathways (e.g., ABA, ethylene) that fine-tune stress-responsive gene expression [182,195]. Under combined metal exposure (e.g., Cd + Pb), crosstalk among these TFs integrates overlapping yet distinct signaling networks, synchronizing antioxidant defenses, metal chelation, and ion homeostasis [194,195,196,197,198,199]. This integrated regulation highlights how plants orchestrate both metal-specific and multi-metal tolerance strategies, providing mechanistic insights valuable for engineering crops with enhanced resilience to HM stress.

Transgenic approaches have proven effective in improving plant tolerance HM stress. Overexpression of genes such as phytochelatin synthase or metal transporters (*HMA3, Nramp*) enhances HM sequestration and limits translocation to shoots, reducing cytosolic toxicity [185,186,187,188,189,190,191]. Antioxidant genes further mitigate oxidative damage and preserve photosynthetic efficiency [158,159,160,165,166,167,168]. Combining these genetic strategies with nanoparticle-assisted remediation offers synergistic benefits: nanoparticles immobilize HMs in the rhizosphere, while transgenic plants efficiently chelate and compartmentalize metals intracellularly. This integrated approach enhances plant growth, stress tolerance, and HM removal efficiency, highlighting the potential of biotechnology in conjunction with nanotechnology for HM-contaminated soils.

CRISPR/Cas9 genome editing offers a precise approach to improve plant tolerance to HM stress. By targeting key genes, transcription factors (e.g., WRKY, MYB, bZIP), and metal transporters, CRISPR/Cas9 can reduce metal uptake and translocation, enhance antioxidant defenses, and strengthen PC or MT-mediated detoxification [182,194]. Studies in rice, Arabidopsis, and wheat have shown that such modifications can improve growth, stress tolerance, and yield under HM exposure, providing an efficient strategy to develop HM-resilient crops.

## 6. Nanoparticle-Mediated Alleviation of Heavy Metal Stresses in Plants

Nanoparticles (NPs) have demonstrated significant potential in alleviating the adverse effects of HM stress in plants [205]. At low concentrations, NPs can stimulate plant growth, enhance stress resistance, and improve photosynthetic efficiency. However, excessive exposure may cause phytotoxicity, including oxidative damage, growth inhibition, and impaired nutrient uptake [206]. Understanding the dose-dependent behavior of NPs is therefore critical for their safe and effective application. NPs can affect HM root-to-shoot translocation by immobilizing metals in the rhizosphere or binding them within root tissues, thus reducing their translocation to shoots [207,208]. For example, Fe_3_O_4_ NPs reduce Cd mobility in wheat roots and shoots, while mercapto-functionalized silica nanoparticles (SiNPs) stabilize Cd in less bioavailable forms [209,210]. ZnO NPs enhance antioxidant defenses in white leadtree (*Leucaena leucocephala*), mitigating Cd and Pb toxicity, and can interact with endogenous phytochelatins and metallothioneins to form stable metal-ligand complexes, further reducing cytosolic toxicity [211,212,213]. These interactions are often concentration-dependent, with optimal NP levels promoting growth and metal detoxification, whereas excessive doses can disrupt cellular redox homeostasis and metal chelation, highlighting the need to determine safe application thresholds. Beyond plant-level effects, the environmental fate of nanoparticles must be considered. Uncontrolled or excessive NP application may affect soil microbial communities, bioaccumulate in non-target organisms, or contaminate soil and water ecosystems. Therefore, careful monitoring, dose optimization, and ecological risk assessment are essential for the sustainable use of nanoparticles in HM remediation. Table 6 presents different nanoparticles used to mitigate HM stress in plants, highlighting their roles in enhancing tolerance and reducing toxicity.

## 7. Nanoparticle-Assisted Bioremediation Against Heavy Metal Stresses

Nanoparticle-assisted bioremediation represents a promising and eco-friendly strategy that merges nanotechnology with biological approaches, particularly the use of microorganisms to detoxify HMs stress [222,223]. For example, bacteria such as *Pseudomonas aeruginosa*, *Bacillus subtilis*, *Rhizobium* spp., and *Staphylococcus aureus* can biosynthesize NPs with high adsorption capacity, enabling effective HM immobilization [224]. Fungal genera like *Aspergillus*, *Rhizopus*, and *Penicillium* contribute to HM remediation [225]. Plant-derived NPs, synthesized using extracts from *Camellia sinensis*, *Citrus limon*, and *Mangifera indica*, benefit from stabilization by secondary metabolites, enhancing their biocompatibility and remediation potential [226]. Polyvinylpyrrolidone (PVP)-coated iron oxide NPs combined with *Halomonas* sp. achieved 100% removal of Cd^2+^ and 80% removal of Pb^2+^ within 24 h, far outperforming NPs used alone [227]. Biogenic manganese oxide (MnO) NPs synthesized by bacteria and fungi have demonstrated excellent cation exchange and redox capacity, enabling effective reduction of Cd, Cu, Pb, and Zn in contaminated soils [228,229]. Remarkably, biogenic MnO from *Pseudomonas putida* MnB1 exhibits 7–8 times greater HM adsorption across diverse environmental conditions. However, the potential ecological risks of nanoparticles should not be overlooked. Excessive or uncontrolled NP application may lead to phytotoxicity, disruption of soil microbial communities, bioaccumulation in non-target organisms, and contamination of water and soil ecosystems. Therefore, careful dose optimization, environmental risk assessment, and monitoring are essential for the safe and sustainable implementation of nanoparticle-assisted remediation strategies. Table 7 summarizes the microbial isolates used for NP biosynthesis and their applications in mitigating HM stress in plants.

## 8. Conclusions and Future Research Directions

Growing prevalence of HM contamination in agricultural ecosystems poses a significant threat to ecosystem health and the sustainability of food production systems. HM stress disrupts essential plant functions by restricting shoot and root development, delaying seedling establishment, while impairing key physiological and metabolic processes including photosynthesis, respiration, water and nutrient uptake, enzymatic activities, and hormonal signaling pathways. These disruptions lead to the excessive generation of ROS, resulting in oxidative stress that causes cellular damage, growth inhibition, and substantial reductions in crop yield and quality. To counteract HM toxicity, plants have evolved a complex network of defense mechanisms operating at physiological, biochemical, and molecular levels. Physiological strategies include root exudation, modulation of ion transport, and regulation of stomatal conductance to limit metal uptake and mitigate stress impacts. Biochemical responses to HM stress involve the synthesis of phytochelatins and metallothioneins for metal chelation and sequestration, alongside the activation of enzymatic (e.g., SOD, CAT, POD) and non-enzymatic (e.g., AsA, GSH) antioxidant systems to scavenge ROS and maintain cellular redox balance. At the molecular level, plants employ transcriptional reprogramming, signal transduction cascades that regulate stress-responsive genes and pathways. Key transcription factors, including WRKY, MYB, bZIP, and NAC families, along with MAPK signaling cascades, play pivotal roles in orchestrating gene expression and enhancing plant resilience under HM stress. Moreover, the advent of engineered NPs and nanoparticle-assisted bioremediation offers promising avenues for enhancing HM immobilization, reducing bioavailability, and promoting safer crop production. Nanoparticle-assisted bioremediation, which integrates biologically synthesized nanoparticles with traditional bioremediation approaches, holds promise for increasing reactivity, improving microbial resilience, and targeting specific contaminants. To bridge existing gaps and advance sustainable HM remediation strategies, future research should focus on:i.Deciphering complex plant–microbiome–NPs interactions to optimize rhizosphere processes for HM detoxification.ii.Applying synthetic biology and CRISPR/Cas9-based gene editing to enhance key regulatory genes, transcription factors, and transporters for improved HM stress tolerance.iii.Integrating multi-omics tools to unravel the regulatory networks and crosstalk between physiological, biochemical, and molecular pathways involved in HM stress tolerance.iv.Evaluating the long-term ecological risks and field performance of NPs, with emphasis on safe design, environmental fate, and regulatory frameworks.v.Developing scalable, field-applicable nanoparticle-assisted phytoremediation protocols that combine engineered plants, beneficial microbes, and smart nanomaterials for site-specific remediation.

Collectively, these efforts will help transform laboratory advances into practical, sustainable solutions for managing HM stress in agroecosystems, ensuring resilient food production systems and safeguarding the environment for future generations.

## Figures and Tables

**Figure 1 plants-14-03834-f001:**
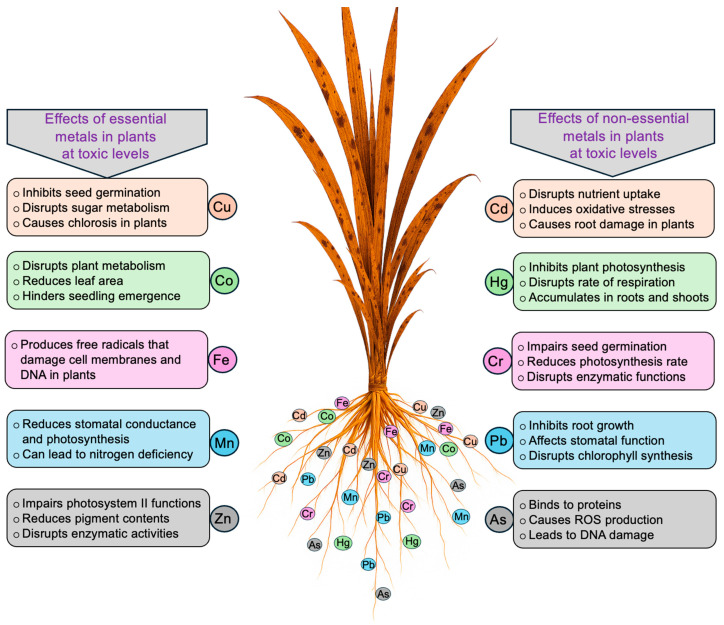
Impact of essential and nonessential metals on plants. Toxicity levels are presented in Table 1. As, arsenic; Cd, cadmium; Co, cobalt; Cr, chromium; Cu, copper; Fe, iron; Hg, mercury; Mn, manganese; Pb, lead; Zn, zinc. Figure created using Adobe Photoshop (Adobe Systems, San Jose, CA, USA).

**Figure 2 plants-14-03834-f002:**
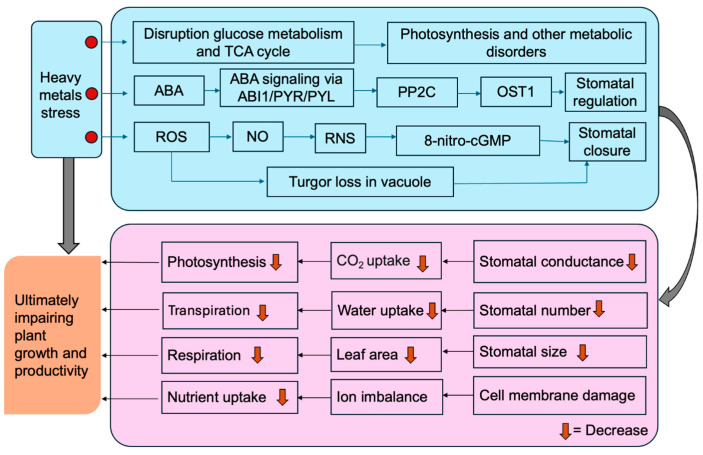
Effects of heavy metal (HM) stresses on physiological functions of plants. HM exposure disrupts glucose metabolism, TCA cycle, stomatal regulation, and multiple signaling pathways, ultimately impairing plant growth and productivity. Blue boxes represent upstream biochemical and signaling disruptions triggered by HM exposure, while pink boxes indicate downstream physiological and growth-related impairments. Horizontal arrows indicate causal relationships or signaling pathways among components. Curved arrows represent feedback effects on stomatal regulation. ABA, abscisic acid; ABI1/PYR/PYL, ABA receptors (ABA insensitive 1/pyrabactin resistance/PYR1-like); cGMP, cyclic guanosine monophosphate; NO, nitric oxide; PP2C, protein phosphatase 2C; OST1, Open Stomata 1; RNS, reactive nitrogen species; ROS, reactive oxygen species; TCA, tricarboxylic acid; 
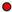
, heavy metals. Figure created using Adobe Photoshop (Adobe Systems, San Jose, CA, USA).

**Figure 3 plants-14-03834-f003:**
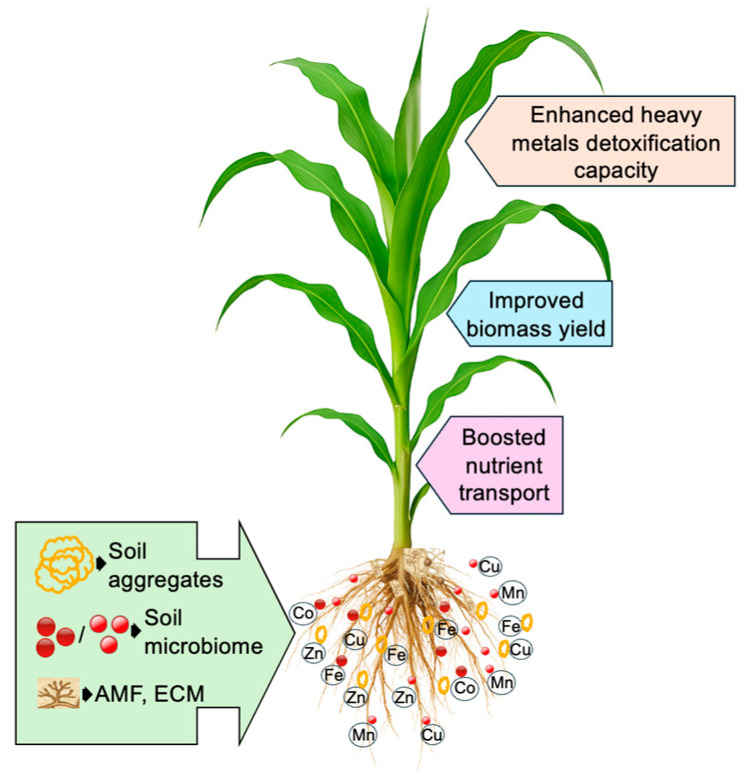
An overview of heavy metals detoxification, nutrient transport, and biomass accumulation in plants by microbial functions. AMF, arbuscular mycorrhizal fungi; Co, cobalt; Cu, copper; ECM, ectomycorrhizal fungi; Fe, iron; Mn, manganese; Pb, lead; Zn, zinc. Figure created using Adobe Photoshop (Adobe Systems, San Jose, CA, USA).

**Figure 4 plants-14-03834-f004:**
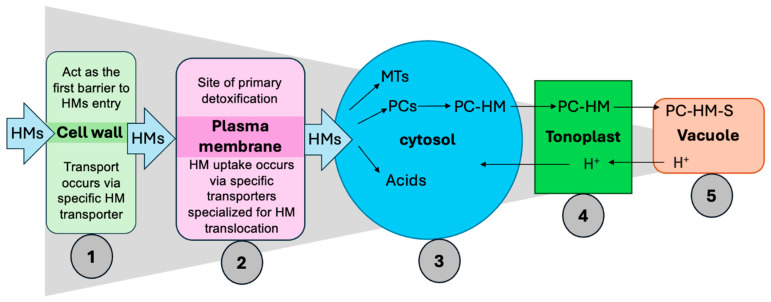
Cellular mechanisms of heavy metals detoxification and sequestration in plants. HMs first interact with the cell wall (1), followed by regulated transport across the plasma membrane (2) into the cytosol (3), where they are chelated by phytochelatins and metallothioneins. The resulting PC-HM complexes are then transported across the tonoplast (4) into the vacuole (5) for sequestration and detoxification, utilizing proton gradients to drive transport and stabilizing HMs in less toxic forms within the vacuole. HMs, heavy metals; MTs, metallothioneins; PCs, phytochelatins; PC-HM, phytochelatin-heavy metal complex; PC-HM-S, stabilized phytochelatin-heavy metal complex. Figure created using Adobe Photoshop (Adobe Systems, San Jose, CA, USA).

**Table 1 plants-14-03834-t001:** Roles and adequate/toxic levels of heavy metals in plants.

Metals	Essentiality and Roles in Plants	Adequate/Beneficial Levels (mg kg^−1^ DW)	Toxic Levels (mg kg^−1^ DW)	References
Arsenic (As)	Non-essential toxic heavy metal	–	≥5	[60]
Cadmium (Cd)		–	≥5	[60,61]
Chromium (Cr)		–	≥5	[62]
Mercury (Hg)		–	≥1	[60,63]
Lead (Pb)		–	≥30	[62,64]
Copper (Cu)	Essential micronutrient	5–30	>30	[65,66]
Iron (Fe)		50–250	>500	[65]
Manganese (Mn)		20–500	>1000	[65]
Molybdenum (Mo)		0.1–5	>10	[65]
Zinc (Zn)		25–150	>150	[67,68]
Nickel (Ni)		0.1–10	>50	[69,70]
Cobalt (Co)	Beneficial for nitrogen fixation in legumes	0.02–2	>10	[65]

Adequate/beneficial levels indicate concentrations typically required for normal plant growth. Toxic levels indicate concentrations at which plants exhibit growth inhibition, chlorosis, or other toxicity symptoms.

**Table 2 plants-14-03834-t002:** Morphological, physiological, and biochemical responses of plants to heavy metal stresses.

Plants	HM Stresses	Morphological, Physiological, and Biochemical Responses	References
Rice (*Oryza sativa*)	Cd	Reduces catalase activity, which impairs H_2_O_2_ scavenging, resulting in higher lipid peroxide levels	[84]
	Ni	Increases the level of H_2_O_2_ and TBARS	[85]
	Pb	Increases lipid peroxidation	[86]
	Hg	Decreases canopy height, tillers number, panicle length, and yield	[87]
Wheat (*Triticum aestivum*)	Ni	Increases electrolyte leakage and lipid peroxidation	[88]
	Cd	Reduces shoot and root development	[89]
Maize (*Zea mays*)	Zn and Ni	Intensifies lipid peroxidation and decrease in permeability of cell membranes	[90]
Sunflower (*Helianthus annuus*)	Cr	Increases lipid peroxidation by stimulating production of malondialdehyde and H_2_O_2_	[91]
	Ni	Inhibits mobilization of stored proteins and amino acids; reduces α-amylase and protease activity	[92]
Lentil (*Lens culinaris*)	Cu	Increases lipid peroxidation in roots	[93]
Pigeon pea (*Cajanus cajan*)	Cd and Ni	Reduces photosynthetic activity	[94]
Siris tree (*Albizia lebbeck*)	Cd and Pb	Cd impairs seedling development and elongation; Pb disrupts stored food material and reduces germination rate	[95]
Indian mustard (*Brassica juncea*)	Cd	Reduces shoot and root biomass and decreases total chlorophyll content in the leaves	[96]
*Silene compecta* and *Thalpsi ochrolucum*	Cu	Damages the electron transport chain involved in photosynthesis	[97]
Thale cress (*Arabidopsis thaliana*)	Cd	Increases lipid peroxidation	[98]
Duckweed (*Lemna minor*)	Cu		[99]

Cd, cadmium; Cr, chromium; Cu, copper; HM, heavy metal; Hg, mercury; H_2_O_2_, hydrogen peroxide; Ni, nickel; Pb, lead; TBARS, thiobarbituric acid reactive substances; Zn, zinc.

**Table 3 plants-14-03834-t003:** Responses of antioxidant enzymes in plants exposed to heavy metal stresses.

Plants	HM	Antioxidant Enzymes Response	References
Maize (*Zea mays*)	Cd	Increases APX and GPX activities	[162]
	Zn	SOD and POD activities increase, while CAT activity decreases at higher Zn levels	[163]
Barley (*Hordeum vulgare*)	Cd	Increases activities of APX and GPX	[164]
Rice (*Oryza sativa*)	Pb	Elevates guaiacol peroxidase, SOD, and GR activities	[86]
Mung bean (*Vigna radiata*)	Cr	APX activity increases, which helps reduce H_2_O_2_ accumulation	[169]
Tomato (*Lycopersicon esculentum*)	Cu	Increases the activities of SOD, POD, and CAT	[170]
Indian mustard (*Brassica juncea*)	Zn	Increases CAT activity, which scavenges H_2_O_2_ and reduces oxidative stress	[171]
Okra (*Abelmoschus esculentus*)	Hg	Increases SOD, APX, and GR activities and decreases CAT activity	[172]
Peregrina (*Jatropha integerrima*)	Zn	POD and CAT activities increase with Zn concentration	[173]
Coffee (*Coffea arabica*)		Increases GR activity, which supports GSH levels for PC biosynthesis	[174]
Water hyacinth (*Eichhornia crassipes*)		CAT activity increases with Ag, Cd, Cr, Pb, and Cu	[175]
Camelthorn (*Alhagi camelorum*)	Cu	Induces PC synthesis and depletes total GSH activity	[176]

Ag, silver; APX, ascorbate peroxidase; CAT, catalase; Cd, cadmium; Cr, chromium; Cu, copper; GPX, glutathione peroxidase; GR, glutathione reductase; GSH, reduced glutathione; HM, heavy metal; Hg, mercury; H_2_O_2_, hydrogen peroxide; PC, phytochelatins; Pb, lead; POD, peroxidase; SOD, superoxide dismutase; Zn, zinc.

**Table 4 plants-14-03834-t004:** Key signaling pathways and molecular components involved in heavy metal stress responses in plants.

Signaling Pathways	Key Components	Heavy Metals	Responses	Genes Involved	References
Calcium-dependent signaling	Ca^2+^ channels, calmodulins (CaMs), calmodulin-like proteins (CMLs), calcium-dependent protein kinases (CDPKs), calcineurin B-like proteins (CBLs)/CBL-interacting protein kinases (CIPKs)	Cr, As, Pb, Cu	Calcium influx triggers antioxidant enzyme activation (e.g., SOD, APX); regulates redox homeostasis; CDPKs and CaMs modulate downstream responses	*AtCBL1*, *CDPK*-like *kinases*, *CAMs*	[178,179,180]
MAPK cascade signaling	MAPKKK → MAPKK → MAPK (MPKs)	Cd, Cu, As, Cr	Phosphorylation of TFs (WRKY, DREB, bZIP, MYB); modulation of stress-responsive genes; interaction with HSPs for defense	*OsMAPK2*, *ZmMPK3/6*, *WRKY*, *ERF*, *bZIP*, *MYB*	[177,181,182]
ROS signaling	ROS (O^2–^, H_2_O_2_, OH^−^), antioxidant enzymes (SOD, CAT, APX), thiol metabolism enzymes	Cd, Cr, As, Pb	Low ROS levels act as signaling molecules; high ROS induce PCD; upregulation of antioxidant genes maintains ROS balance	*OsGSTL2*, *OsMATE1/2*, *DHAR*, *GR*, *SOD*, *CAT*	[178,183]
Hormonal Signaling	ABA, JA, ET, SA, EIN2/3, JAZ, AP2/ERF transcription factors	Cd, Cr, As	Phytohormones regulate transcription and crosstalk with MAPK cascades; influence root development and HM detoxification	*AP2/ERF*, *ACS*, *OsARM1*, *AtMYB*, *AB15*, *TGAL3*	[180,184]
Crosstalk and integration	Interactions among Ca^2+^, ROS, MAPKs, hormones, nitric oxide	Cd, Pb, As, Cr, Cu	Synergistic and antagonistic interactions among signaling pathways coordinate stress responses; modulate TF networks	Multigene families: *WRKY*, *bZIP*, *HSF*, *MYB*, *ERF*	[177,178]

ABA, abscisic acid; AP2/ERF, APETALA2/ethylene-responsive factor; APX, ascorbate peroxidase; As, arsenic; Ca^2+^, calcium ion; CAT, catalase; Cd, cadmium; CBL, calcineurin B-like protein; CDPK, calcium-dependent protein kinase; CIPK, CBL-interacting protein kinase; Cr, chromium; Cu, copper; DHAR, dehydroascorbate reductase; ET, ethylene; GR, glutathione reductase; H_2_O_2_, hydrogen peroxide; HSP, heat shock protein; JA, jasmonic acid; MAPK, mitogen-activated protein kinase; MAPKK, MAPK kinase; MAPKKK, MAPK kinase kinase; OH^−^, hydroxyl radical; O_2_^−^, superoxide anion; PCD, programmed cell death; ROS, reactive oxygen species; SA, salicylic acid; SOD, superoxide dismutase; TF, transcription factor; Zn, zinc.

**Table 5 plants-14-03834-t005:** Roles of transcription factors in plant adaptation to heavy metal stresses.

Plants	TFs	Gene(s)	Key Findings	References
Wheat (*Triticum aestivum*)	HSF	*TaHsfA4a*	Upregulates metallothionein genes under Cd stress	[194]
Rice (*Oryza sativa*)	MYB	*OsMYB45*	Downregulation increases Cd sensitivity; regulates antioxidant activity	[182,195]
	bZIP	-	Involved in auxin and HM signaling crosstalk	
	WRKY	-	Activated by MAPK pathways under HM stress	
Sorghum (*Sorghum bicolor*)	MYB	*SbMYB15*	Confers Cd and Ni stress tolerance	[196]
Walnut (*Juglans regia*)	MYB	*JrMYB2*	Improves tolerance to Cd stress	[197]
Tomato (*Solanum lycopersicum*)	HSF	*HSF1A*	Induces melatonin biosynthesis for Cd tolerance	[198]
Rapeseed (*Brassica napus*)	bZIP	*BnbZIP2*, *BnbZIP3*	Upregulated under drought and Cd; involved in stress signaling	[199]
Arabidopsis (*Arabidopsis thaliana*)	MYB	*AtMYB4*	Improves antioxidant defense under Cd stress	[200]
	WRKY	*AtWRKY12*	Downregulated under Cd; represses *GSH1* to negatively regulate Cd tolerance	[201]
	WRKY	*AtWRKY13*	Upregulated under Cd; activates *PDR8* to positively regulate Cd tolerance	[202]
	WRKY	*WRKY33*	Regulates HM uptake via *IRT1* regulation under Cd stress	[203]
	bZIP	*AB15*	Interacts with *MYB49* to reduce Cd uptake via *IRT1* inactivation	[204]

Cd, cadmium; Ni, nickel; GSH1, glutathione synthetase 1; HM, heavy metal; PDR8, pleiotropic drug resistance 8; IRT1, iron-regulated transporter 1; MAPK, mitogen-activated protein kinase; HSF, heat shock transcription factor; MYB, myeloblastosis transcription factor; TFs, transcription factors; WRKY, tryptophan-arginine-lysine-tyrosine transcription factor; bZIP, basic leucine zipper transcription factor.

**Table 6 plants-14-03834-t006:** Nanoparticles for heavy metal stress mitigation in plants.

Nanoparticles	Plant Species	HMs	Reduction of HMs (%)	Key Findings	References
Zinc oxide (ZnO)	Rice, fenugreek, and *Leucaena leucocephala*	Pb, Cd, Cr, Cu	Pb: 79–85; Cd: 80–87; Cr: 38–81; Cu: 60	Improves growth and Zn uptake; reduces HM accumulation	[213,214]
Cerium oxide (CeO_2_)	Rice	Cd	8.4	Reduces growth inhibition and oxidative stress	[152]
Astaxanthin-functionalized gold (Ast-Au) NPs			26–86	Enhances chlorophyll content and amino acid metabolism; scavenges ROS	[215]
Titanium dioxide (TiO_2_)	Rice and cucumber	Pb, As, Al	34–97	Reduces HM contamination and toxicity	[216]
Iron oxide (Fe_3_O_4_)	Wheat	Pb, Zn, Cd, Cu	Roots: 24–68; Shoots: 11–100	Reduces oxidative stress and growth suppression	[217,218]
Selenium NPs (Se, Bio-Se)	Coriander and pak choi	Cd, Pb, Hg	Cd: 21–31; Pb: 5–30; Hg: 3–23	Enhances antioxidant defense; reduces HM uptake	[219,220]
Graphene oxide	Lettuce	Cd	-	Reduces Cd toxicity; improves photosynthesis, chlorophyll content, antioxidant enzymes, and biomass	[221]

Al, aluminum; As, arsenic; Bio-Se, biologically synthesized selenium; Cd, cadmium; Cr, chromium; Cu, copper; Fe, iron; HM, heavy metal; Hg, mercury; NPs, nanoparticles; Pb, lead; ROS, reactive oxygen species; Zn, zinc.

**Table 7 plants-14-03834-t007:** Microbial and plant-based biosynthesis of nanoparticles and their role in mitigating heavy metal phytotoxicity.

Nanoparticles	Microorganisms/Plant Extracts Used	Alleviation of Phytotoxicity/Key Findings	References
Silver NPs	*Escherichia coli*	Rapid reduction of Ag^+^ ions within minutes	[230]
	*Pseudomonas stutzeri*	Silver-resistant; accumulates silver and reduces its toxicity	[231]
	*Solanum xanthocarpum* (berry extract)	Enhances Ag^+^ ion reduction rate via phytochemicals	[232]
	*Convolvulus arvensis* (leaf extract)	Achieves 98.99% Cu^2+^ ion removal via adsorption	[233]
Gold NPs	*Bacillus subtilis*	Acts as a biocontrol agent with antifungal properties	[234]
	*Aspergillus japonicus*	Reduces Au(III) to Au(0); immobilized AuNPs	[235]
	*Colletotrichum* sp.	Reduces and caps gold NPs	[236]
ZnO NPs	Green algae	Converts metal ions into zero-valent metals via phytochemicals	[237]
	*Citrus limon* (leaf extract)	Non-toxic synthesis; biomolecule-rich extract enhances safety	[238]
Lead NPs	*Clostridium pasteurianum*	Reduces Cr(VI) to Cr(III); ~70% remediation efficiency	[239]
Iron NPs	*Chlorococcum* (alga)	Biosynthesized Fe NPs removed 92% Cr vs. 25% by bulk Fe	[240]
	*Aspergillus oryzae*	Cost-effective and eco-friendly NP synthesis for remediation	[241]
TiO_2_ NPs	*Aspergillus niger*	Reduces Cr(VI) toxicity and DNA damage in *Helianthus annuus* by minimizing total Cr uptake	[242]
Biogenic Fe–Mn oxides (BFMO)	*Pseudomonas* sp.	Converts As(III) to less mobile As(V); enhances arsenic remediation	[243]
CdS NPs	*P. aeruginosa*	EPS-enriched CdS NPs enhance cadmium ion adsorption and stabilization	[244]

Ag^+^, silver ions; As, arsenic; As(III), arsenite; As(V), arsenate; AuNPs, gold nanoparticles; CdS, cadmium sulfide; Cr, chromium; Cr(III), trivalent chromium; Cr(VI), hexavalent chromium; EPS, extracellular polymeric substances; Fe–Mn, iron–manganese; NPs, nanoparticles; TiO_2_, titanium dioxide.

## Data Availability

No new data were created or analyzed in this study. Data sharing is not applicable to this article.

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
