# Peer review of "Plant Responses to Heavy Metal Stresses: Mechanisms, Defense Strategies, and Nanoparticle-Assisted Remediation"

_plants, 2025, doi:10.3390/plants14243834_

Round 1
Reviewer 1 Report (Previous Reviewer 2)
Comments and Suggestions for Authors
The authors have satisfactorily responded to all comments, and the revised manuscript meets the required standards for publication in Plants.
Reviewer 2 Report (Previous Reviewer 1)
Comments and Suggestions for Authors
After reviewing the revised manuscript “Plant Responses to Heavy Metal Stress: Mechanisms, Defense Strategies, and Nano-Assisted Remediation”, I have no further comments and recommend acceptance for publication in MDPI Plants.
I thank the authors for their clear and constructive responses.
Reviewer 3 Report (Previous Reviewer 4)
Comments and Suggestions for Authors
I appreciate the revised version of the manuscript. All points revealed in my first review were now improved.
This manuscript is a resubmission of an earlier submission. The following is a list of the peer review reports and author responses from that submission.
Round 1
Reviewer 1 Report
Comments and Suggestions for Authors
After reading the manuscript “Plant Responses to Heavy Metal Stress: Mechanisms, Defense Strategies, and Nano-Assisted Remediation”, I have a few comments, namely:
- Line 155: "Zia mays" should be corrected to Zea mays.
- Chapter 5.1: The authors write that symbiosis with AMF is a critical plant defence strategy in environments contaminated with HMs. My question is whether HMs translocate from the roots to other parts of the plant, or whether they remain in the root tissues. If they do move, do they translocate at lower concentrations? In the summary of this chapter, I would like to see information on whether the scientific literature addresses this issue or whether the mechanisms are still unknown. The authors do point out that AMF can retain metals in the root tissues through chelation. Another important contribution of AMF is the secretion of glomalin, a metal-binding glycoprotein that plays a key role in HM inactivation in the soil. However, this process varies between plant species, and soils are not always rich in arbuscular mycorrhizal fungi, despite their essential role in enhancing plant tolerance to HMs by forming symbiotic associations with roots.
Furthermore, in the first sentence of this chapter, the authors mention mycorrhizal fungi, specifically arbuscular mycorrhizal fungi (AMF) and ectomycorrhizal fungi, as playing an essential role in increasing plant tolerance to HMs by forming symbiotic associations with roots. However, nothing is stated about the latter group: why they are not discussed, or whether they are considered less important in HM protection. It may be worth including a discussion of ectomycorrhizal fungi as well.
I also find this chapter missing explicit statements on how immobilisation by mycorrhizal associations enhances biomass yield, improves the ability to detoxify HMs during the high-growth phase, and increases nutrient transport in plants. This is implied in Figure 3, but it would be useful to state it directly. Overall, this chapter needs slight improvement.
- Line 381: Gene names should be written in italics, as in Tables 3 and 4. Please review the entire manuscript and correct all gene names accordingly.
The comments above are reasonable and constructive. Most involve minor editorial issues, such as correcting the scientific name (Zea mays). However, comment two regarding the AMF chapter requires deeper analysis. This is particularly important because the manuscript does not clarify whether HMs are translocated from roots to shoots, and it overlooks the role of ectomycorrhizal fungi in protection mechanisms.
Regarding comment three, the use of italics for gene names is critical in view of nomenclature standards in molecular biology. I highlighted this issue because there is widespread inconsistency in the literature, which creates confusion between gene and protein references. International nomenclature standards define this distinction clearly:
- Gene names = italics (e.g. SOD1, CAT, WRKY1)
- Protein names = regular font (e.g. SOD1, CAT, WRKY1)
Overall, the paper deserves publication after corrections. It is a solid and comprehensive study with clear scientific merit. Implementing these revisions will significantly improve its quality. Once corrected, the manuscript will serve as a valuable and professional resource for researchers studying phytoremediation and heavy metal stress in plants.

Reviewer 2 Report
Comments and Suggestions for Authors
This manuscript presents a comprehensive and novel investigation into plant responses to heavy metal stress, focusing on physiological, subcellular, and lipidomic mechanisms, as well as nano-assisted remediation strategies. The study is well-designed and provides valuable insights into the defense strategies of plants under heavy metal exposure. However, several issues related to structure, clarity, and depth of discussion should be addressed to enhance the manuscript’s impact and scientific rigor. The authors should clearly highlight the novelty of this review
Lines 23–32: The abstract is informative but lacks quantitative data. Add representative effect sizes from cited studies (e.g., “Si reduced Cd uptake by ~35–60% in rice; SOD activity increase 40–80% under Cu stress”), or at least one percentage change for growth, photosynthesis, or antioxidant responses.
Line 30: “enzymatic and nonenzymatic activity” remove double spaces; clarify as “enzymatic and non-enzymatic antioxidant activity.”
Lines 38–45: Nano-remediation is introduced, but risk and field relevance are underdeveloped. Add 1–2 sentences on transformation/fate (e.g., dissolution, agglomeration), dose–response boundaries, and environmentally relevant concentrations.
Line 54: Replace “HMs frequently detected…” with “HMs are frequently detected…”.
Clearly mention the novelty of the review article and how it differs from previous reviews. https://doi.org/10.1016/j.jhazmat.2023.131039, https://doi.org/10.3390/nano11010026, https://doi.org/10.1007/s41204-022-00230-8, 10.1155/2015/756120
Lines 91-94: Sentence structure unclear.
Lines 103–105: Arsenic subsection briefly distinguish As(III) vs As(V) uptake routes and typical root vs shoot partitioning to add mechanistic depth.
Lines 137-183, Authors describe various biochemical responses to HMs stress, including the synthesis of phytochelatins and metallothioneins for metal sequestration. While Authors mention that these compounds play a role in chelating metals and reducing cytosolic toxicity, could Authors provide more specific examples of how these molecules interact with particular HMs such as cadmium or arsenic? For example, does the metal-binding specificity of phytochelatins vary depending on the HM type, and how does this influence the efficiency of the detoxification process? Additionally, could Authors discuss whether these chelation mechanisms vary across different plant species, and if so, how this affects their overall stress tolerance under different HM concentrations?
Lines 377-386: Authors mention transcription factors (TFs) such as WRKY, MYB, and bZIP in regulating plant responses to HM stress. Could Authors elaborate on how the crosstalk between these TFs is specifically modulated under the presence of multiple heavy metals (e.g., cadmium and lead together) compared to a single-metal exposure? Are there particular TFs that are preferentially activated under specific HM stresses, and how do these differences contribute to a plant's ability to adapt or tolerate multi-metal stress? It would be insightful to provide a comparison between these TFs' roles in individual metal stress versus combined metal stress.
Lines 396-420): In article discussion of nanoparticle (NPs) remediation, Authors mentioned that nanoparticles such as ZnO and Fe3O4 are used to reduce HM toxicity (Lines 396-420). Could Authors elaborate on the dose-dependent effects of nanoparticles in remediating HM stress? For instance, while low concentrations of ZnO nanoparticles promote growth, excessive exposure can lead to phytotoxicity. How do nanoparticles affect plant root-to-shoot metal translocation at varying concentrations, and is there a threshold beyond which the nanoparticles themselves become toxic to plants? Also, how do nanoparticles interact with the plant's own metal-chelating systems like phytochelatins and metallothioneins?
Specify the names of the software or websites used to create the figures.
Reviewer 3 Report
Comments and Suggestions for Authors
This review systematically summarizes the response mechanisms, defense strategies, and nano-assisted remediation technologies of plants under heavy metal stress. It covers comprehensive content with a clear structure, encompassing multi-level responses from physiological and biochemical aspects to molecular mechanisms, thus demonstrating strong comprehensiveness and cutting-edge nature. The manuscript is timely, but still need some revision before acceptance.
It is suggested that "nano-remediation" in the keywords be revised to "nanoparticle-assisted remediation" to more accurately reflect the content of the paper.
In Section 5, which focuses on transcription factors (TFs), Table 4 provides an extensive list of TFs; however, it lacks a mechanistic summary of the functions of key TFs. It is recommended to add a paragraph of integrated analysis.
Although the sections on nano-remediation (Sections 6–7) present novel content, the discussion on the potential ecological risks of nanomaterials is insufficient. It is suggested to supplement relevant content.
Several high-quality studies from the past two years (2023–2024) focusing on plant heavy metal signaling pathways or the environmental behavior of nanomaterials could be additionally cited to enhance the timeliness of this review.
The conclusion section provides a good summary of the entire paper. However, content such as "CRISPR/Cas9" mentioned in the future outlook has not been sufficiently elaborated in the main text. It is recommended to make appropriate explanations in the corresponding chapters. The formatting of some references is inconsistent (e.g., mixed use of abbreviated and full names for journal titles), and it is recommended to standardize them in accordance with the journal's requirements.
Reviewer 4 Report
Comments and Suggestions for Authors
The manuscript ID: plants-3856153 entitled ”Plant Responses to Heavy Metal Stress: Mechanisms, Defense 2 Strategies, and Nano-Assisted Remediation’’ by Jarin et al. report the morphological, physiological, biochemical, and molecular responses of plants to HM stress, focusing on their underlying defense mechanisms and current strategies in phytoremediation and nano-remediation. I consider that the review paper contains an update in the effect of HM in plants and many information to advance HM remediation. The review is consolidated with many schematic figures and tables. Below you can find to be considered the relevant points:
Major points
- In the abstract, I suggest to delete the sentence ‘’ This review explores the structural, functional, biochemical, and genetic mechanisms underlying plant responses to HM stress.’’ Or include it at the end of abstract.
- In the keywords, ‘’ plant tolerance mechanisms’’ is very general. I suggest to replace it by two others keywords such as ‘’oxidative stress’’ and ‘’Signaling pathways’’.
- In the Figure 1 and in the text of section 2, the authors didn’t define ‘’low and high concentrations of HM’’ and if it depend in the nature of the metal.
- Line 140: what is the meaning of ‘’ intra-leaf CO2 levels’’?
- I’m not convinced with the role of HSP in response to HM stress. I suggest to delete this paragraph from Line 330 to line 339 or more detailed this aspect.
- This review lack one section related to functional study such as some data in relation with transgenic plants that overexpressed genes that promote tolerance to HM stress. It will be nice to see the biotechnological aspect with the nanoparticle’s strategies.
- References should be updated mainly in the case of review paper. There are many references from 1990 to 2015, more than 20 years ago.
- References list should be checked. The name of the journal should be presented in abbreviation. In all cases, refer to the journal instructions.
Minor points
- Line 69, change ‘’ heavy metals’’ by ‘’HM’’. The same remark in line 129.
- Line 155, change ‘’ (Zia mays)’’ to ‘’ (Zea mays)’’.
